# Occupational Infection Risk with Multidrug-Resistant Organisms in Health Personnel—A Systematic Review

**DOI:** 10.3390/ijerph16111983

**Published:** 2019-06-04

**Authors:** Claudia Peters, Madeleine Dulon, Albert Nienhaus, Anja Schablon

**Affiliations:** 1Competence Centre for Epidemiology and Health Services Research for Healthcare Professionals (CVcare), University Medical Centre Hamburg-Eppendorf (UKE), 20246 Hamburg, Germany; albert.nienhaus@bgw-online.de (A.N.); a.schablon@uke.de (A.S.); 2Department of Occupational Medicine, Public health and Hazardous Substances, Institution for Statutory Accident Insurance and Prevention in the Health and Welfare Services, 22089 Hamburg, Germany; madeleine.dulon@bgw-online.de

**Keywords:** health personnel, multidrug-resistant organisms, colonization, occupational exposure

## Abstract

The increase in multi-drug-resistant organisms (MDROs) in the last years has become a public health problem. MDROs are partially responsible for numerous nosocomial infections, extended hospital stays, high costs, and high mortality. In addition to methicillin-resistant *Staphylococcus aureus* (MRSA) and vancomycin-resistant enterococci (VRE), Gram-negative bacteria are also a key area of focus. The knowledge of MDROs among the medical staff in the occupational context is limited, with the exception of MRSA. Therefore, a systematic review was carried out to determine the occupational risk for employees posed by MDROs. The search included studies from the year 2000 onwards among personnel who had contact with MDROs. A total of 22 primarily cross-sectional studies in hospital or geriatric care settings were found, with large differences regarding number of participants, examination method, inclusion of a control group, and study quality. The most frequently examined pathogens were extended-spectrum ß-lactamase (ESBL)-producing bacteria with a prevalence of 2.6–48.5%, VRE (0–9.6%), and MRSA (0.9–14.5%). There are only few qualitatively good studies available on MDROs’ risk infection for employees in the health service. Any comparison of the results was limited by data heterogeneity. More research is required to describe the occupational risk of infection with MDROs.

## 1. Introduction

While in recent decades Gram-positive methicillin-resistant *Staphylococcus aureus* (MRSA) has been the main subject of interest, the increased occurrence of vancomycin-resistant enterococci (VRE) and multidrug-resistant Gram-negative bacteria (MDR-GNB) is now coming into focus. These pathogens represent a major health risk due to their adaptability and the development of resistance. As a result, they are associated with limited or no therapies available, prolonged treatment times, high costs, and high mortality rates [1,2,3]. 

The situation with MRSA is best researched because this pathogen has presented the greatest problem worldwide in the past few decades. MRSA diffusion has been slowly decreasing in Europe in recent years, while the proportion of VRE cases has risen significantly [4,5]. This trend can also be observed in Germany, although prevalence rates of blood stream infections vary by region [6].

MRSA prevalence is stated at 0.7% for the general population in Germany [7]. The frequency of MRSA colonization among patients in the various healthcare fields was found to be between 1% and 24% in Europe [8]. Among health personnel, the average prevalence rates were 4.6% [9] and 5% [10]. One review of MRSA studies in non-outbreak settings showed prevalences between 0.2 and 15% [11]. For Germany, studies among employees in healthcare institutions showed MRSA prevalence rates of 0.4 to 4.5% [12], while for geriatric care and patient transport services, these rates were between 0 and 3.2% [13,14,15,16,17]. 

The scientific knowledge of colonization by VRE or MDR-GNB in employees is limited, although studies in patients [5,18,19] and residents of nursing homes [13,20] show colonizations with multi-drug-resistant organisms (MDROs) in health personnel. In nosocomial outbreak situations, patients and staff are often screened in order to identify the source. In one review, Ulrich et al. [21] showed that staff were rarely responsible for the outbreak. However, studies of non-outbreak situations are required to assess the occupational infection risk in health personnel. A systematic literature search was therefore conducted in order to clarify whether employees in the health service have an increased risk of infection with MDROs in non-outbreak situations.

## 2. Materials and Methods 

In order to address the research question, a systematic review in accordance with the PRISMA statement was conducted [22]. 

### 2.1. Inclusion and Exclusion Criteria

Inclusion was determined by the PEO criteria (Population, Exposure, Outcome) [23]. Only those studies in which health personnel (=population) had direct (nurses or doctors) or indirect patient contact (=exposure) through handling infected material or an infected environment (laboratory, cleaning staff) and had been examined for multidrug-resistant organisms (=outcome) were considered. Included were all epidemiological studies published from 2000 onwards that examined the occupational exposure of health personnel to MDROs. There were no regional restrictions. Exclusion criteria included private transmission, outbreaks, multidrug-resistant tuberculosis, and hand or environment screening alone. Furthermore, no publications concerning isolated MRSA testing were included because numerous publications are available on this topic [9,10,11,24]. 

### 2.2. Database Search

A literature search was conducted in PubMed, MEDLINE/Ovid, Web of Science, and CINAHL databases and supplemented by a manual search in the reference lists of the researched publications. The last update was conducted on 28 March 2019. The following search strategy was applied for PubMed: 

health personnel OR health professional* OR healthcare personnel OR healthcare worker* OR healthcare professional* OR occupational exposure OR occupational risk* OR occupational transmission* AND multidrug resist* OR MDRO* OR vancomycin-resistant OR enterococ* OR VRE* OR gram-negative bac* OR GNB* OR extended-spectrum b-lactamas* OR ESBL* OR carbapenemase

### 2.3. Study Selection

After duplicate removal, the titles and abstracts were screened. Suitable hits were checked independently by two reviewers (CP, AS) with regard to the inclusion criteria. In case of any divergence, an additional reviewer was consulted (MD), and a consensus reached through discussion. Data extraction and quality assessment of the included studies were carried out using a standardized data sheet. The data extraction took into account the study design, study region, study period, setting, sample size, sample type, MDRO species investigated, and their prevalence. The quality assessment criteria were developed and adapted using evaluated checklists [25,26]. The study quality was evaluated using the following characteristics: inclusion criteria, setting, sample collection and microbiological method, adequate sample size, statistical method, discussion of limitations, and inclusion of a comparison group. Where six to seven criteria were fulfilled, the study quality was assessed as high, four to five criteria signified moderate quality, and fewer than four criteria signified low quality.

## 3. Results

A total of 3052 studies were identified using a systematic search of the databases. After removing duplicates, 1029 hits were checked for suitability, with 67 of these included in full-text screening. In total, 22 publications met the inclusion criteria and were included in the analysis (Figure 1). Of these studies, 11 were conducted in Europe, 4 in North America, and the remaining in other countries around the world (South America, Asia, Africa). The most commonly used study design was a cross-sectional study (*n* = 17). The study period varied significantly between one and several years. The majority of studies were carried out in a hospital setting (*n* = 16); in other studies, employees in geriatric care settings were included as the study population. The study scope indicated a variation of 13 to 1185 people. Ten studies included much fewer than 100 participants. In three studies, a control group was included that comprised non-medical staff or staff without patient contact. Two studies [27,28] fulfilled all seven criteria for the quality assessment. Nine studies were categorized as moderate-quality and 10 as low-quality. A review [29] was not included in this evaluation.

### 3.1. MDROs

Various organisms were investigated in the individual studies. For Gram-negative bacteria, the prime focus was on extended-spectrum ß-lactamase (ESBL) producers. ESBL producers were most commonly analyzed using rectal or faecal samples. In a total of 13 studies, ESBL producers’ prevalence varied between 2.6% and 48.5%. For the hospital sector, the results are shown in Figure 2. In comparison to staff without patient contact, the study by Decker et al. [27] showed an ESBL frequency of 4.0% versus 2.9% for the medical personnel. In other studies, there was no adequate control group. Carbapenemase-producing organisms (CPO) were analyzed in four studies and did not result in any positive finding in the hospital setting. In geriatric care settings, colonization by MDROs between 0% and 1.5% was described for staff in the review by Aschbacher et al. [29] and was investigated in the study by March et al. [30] using metallo-ß-lactamase (MBL). An AmpC diagnostic was carried out in two studies, finding prevalence rates of 1.5% in geriatric care settings [30] and 3.0% in clinical settings [31]. 

As an additional MDR-GNB, *Acinetobacter baumannii* was investigated in two studies with frequencies of 0 and 0.5% [32,33]. 

Gram-positive bacteria were analyzed using VRE testing in nine studies and showed colonization rates of 0–9.6%, although no VRE was found in most studies. In the study by Baran et al. [34], staff with patient contact showed a prevalence of 9.6%, while no VRE was diagnosed in any other staff without patient contact. MRSA was primarily investigated in studies in a geriatric care setting. The frequency of positive findings here was between 3.1% and 14.5%. Within the clinical setting, the results of two studies showed a prevalence of 0% [35] and 0.9% [28] for MRSA colonisation. 

In four studies, the results were presented generally as related to MDROs with additional differentiation of the species causing multidrug-resistance. *Enterobacteriaceae* [36], *Klebsiella pneumoniae*, *Enterobacter aerogeneses*, *A. baumannii* [28], and *Escherichia coli* [37] were most commonly identified.

### 3.2. Study Description

The studies included in the review were primarily carried out in a hospital setting (*n* = 16). Staff working in geriatric care were included in six studies investigating MDROs. The studies are listed in Table 1 for hospital setting and in Table 2 for geriatric care setting. The studies in both settings are described in more detail below.

#### 3.2.1. Health Personnel in Hospital Settings

The prevalence and risk factors of ESBL-producing *Enterobacteriaceae* were investigated in a prospective study in medical staff and family members of infected patients in rehabilitation centres in Israel, Italy, Spain, and France. Samples were taken via rectal swabs. In a total of 1001 staff members, 3.5% showed a positive result for ESBL. In Spain, the prevalence was the highest at 10.6%, although only 47 staff members were tested. The positive cases primarily pertained to nursing staff and staff with direct patient contact. Family members were much more commonly colonized, with prevalence rate at 9.0% [38]. A prospective study was conducted among Swedish healthcare students who completed a clinical assignment abroad. In addition to a faecal swab before and after their stay abroad, demographic and destination-related details were also recorded. South-East Asia and Africa were the most popular destinations, and students were abroad for an average of 45 days (median). After their travels, 35% had a newly acquired ESBL-producing *E. coli*. The most important risk factors were travelling to South-East Asia and treatment with antibiotics during the trip. There was no statistically significant association between patient contact and colonization status [39]. A retrospective study design was used in Brazil to investigate MDRO prevalence among medical staff with occupational contact to MDROs. The electronic patient files of hospital employees from various sites were evaluated for this purpose. The sample comprised staff who had received inpatient treatment and from whom a biological sample was taken during the first five days. During the study, 105 out of 1487 employees with a hospital stay underwent microbiological testing. No MDROs were identified [40]. 

VRE prevalence in stool samples of hospital employees and their family members was investigated via a cross-sectional study. The participants were divided into households with and without patient contact. VRE was identified in 5 out of 52 employees with patient contact (9.6%). Among family members of these employees, VRE prevalence was also higher than for relatives of employees without patient contact (7.3% versus 2.2%) [34]. Employees and patients from intensive care wards and rehabilitation departments from two New York hospitals were tested for *A. baumannii* using nasal and hand swabs. Of a total of 184 employees, 3.3% tested positive, with one employee showing a multi-resistant strain. The prevalence of MDROs was 8.2% for patients. Four participants (one employee, three patients) showed nasal colonization, while four employees and seven patients showed hand colonization, and both regions were colonized in five participants (one employee, four patients). Predictors among employees were prior skin damage and working in one of the two hospitals [41]. The prevalence of VRE in staff and patients on intensive care wards and in surgical departments was investigated in a Turkish hospital. VRE was not identified in any of the staff members. One patient (1/287) tested positive after being transferred from another hospital [41]. In the paediatric department of a hospital in Madagascar, the prevalence of ESBL-producing *Enterobacteriacea* was measured in staff and patients via rectal swab; the environment was also sampled. Children aged under 15 years of age were tested upon admission and discharge (21.2% versus 57.1% positive). Almost half (48.5%) of the staff tested positive for ESBL producers, primarily *E. coli* or *K. pneumoniae*. The investigation was conducted as a cohort study for patients. For the staff, it appears that a cross-sectional study was carried out, but this is not explicitly reported [42]. 

In a septic orthopaedics department of a Swiss hospital, swabs were taken to investigate staff, patients, and the environment for ESBL-producing *Enterobacteriaceae*. There were six positive cases among the staff (6/41). The typing did not reveal any overlaps with patients’ samples. No other details about the employees were provided [43]. In Egypt, employees from various hospital departments were tested for ESBL and AmpC ß-lactamase via stool samples. Of a total of 200 samples, there was a prevalence of 21% for ESBL- and of 3% for AmpC-producing *E. coli*, of which seven isolates were identified as being multidrug-resistant [31]. The presence of MDROs in the oral cavity was tested in medical staff and service employees in an oncology hospital in Brazil by saliva sampling. Of 294 participants, 18.7% (24% medical versus 13% service staff) were colonized with *Enterobacteriaceae*, with various multidrug-resistant species found in 9.2% [36]. Medical students and patients at a university hospital in Hungary were tested for ESBL by routine stool sample. The prevalence among students was 2.6%, 7.4% for inpatients and 3.1% for outpatients. ESBL-producing *E. coli* was frequently found in the students [18].

Staff from a hospital in Israel were investigated for carbapenemase-producing *Enterobacteriaceae* via rectal swab. No cases of CPO were found, despite 75% of participants having contact with CPO patients. The patient incidence rate at the hospital was stated at 2.6/1000 admissions [44]. Intestinal colonization with MDRO was tested in 74 physicians and further non-medical staff (*n* = 33) via rectal swab in a German study. At the same time, risk factors such as travel, occupation, treatment abroad, vegetarianism, animal contact, and consumption of raw meat were recorded. ESBL-producing *E. coli* was found in 3.7% of the participants. Other MDROs such as MRSA and VRE were not found, and the risk factors did not show any significant impact of colonization by MDROs [35].

In a Chinese study, the molecular biological colonization of the nose and hands was investigated in nurses, doctors, and non-medical staff on intensive care wards (*n* = 1318). The analysis showed 8.8% multi-drug-resistant Gram-negative bacteria in medical staff and 15.0% in non-medical staff. Positive results were much more common for nasal swabs. The most common multidrug-resistances were caused by *K. pneumoniae, E. aerogeneses,* and *A. baumanii* [28]. The extension of the study by one year resulted in an MDR-GNB prevalence of 11.0% for 1,451 participants, 10.7% for medical staff (139/1296), and 12.9% (20/155) for non-medical staff. These results were not included in the review because the publication was issued in Chinese, and only an abstract was available in English [45]. An American cross-sectional study tested intestinal colonization with MDROs in 755 hospital staff with and without patient contact. ESBL producers were found in 3.4% of all participants, with 0.1% of them testing positive for carbapenemase-producing organisms. The group with patient contact primarily comprised nursing staff and physicians, while the controls generally included administrative and other staff. Patient contact as exposure showed practically no differences for MDRO colonization [27]. As part of a surveillance investigation, staff, patients, and the environment at a new US military base in Iraq were tested for MDR-GNB. In order to be able to identify colonizations early, swabs of the axilla and/or groin were taken after patient admission for up to six months after the opening of the facility and, for staff, every two to four weeks. With a total of 246 samples from 80 medical staff, three MDROs (two *E. coli*, one *Achromobacter*) were identified. Among patients, primarily *K. pneumoniae* and *E. coli* were isolated, which were also found in the environmental analysis. An Iraqi nationality was identified as a risk factor for MDR-GNB, with an OR of 2.9 (95% CI 1.3–6.3) [37].

#### 3.2.2. Health Personnel in Geriatric Care Setting

A review by Aschbacher et al. [29] investigated MDRO in residents and staff at Italian long-term care facilities (LTCF) in comparison to other European countries. Three Italian studies, which also included the analysing of staff, took place between 2000 and 2016. In one study, only MRSA were examined, while two others also tested for ESBL, CPO, and VRE. There was no evidence of VRE, however. The two studies are presented below [30,46]. A point prevalence study in Bolzano, Italy, investigated the prevalence of MDROs among residents of LTCFs, a geriatric department of a hospital, and among staff. Nasal, oropharyngeal, inguinal, and rectal swabs were taken for this purpose, as well as urine samples. For the 69 members of the staff, there was an overall prevalence of 27.5% for MDROs: 14.5% each for MRSA and ESBL-producing *Enterobacteriaceae*. No cases of VRE were found. The combined results for patients and residents were 29.5% for MRSA, 48.1% for ESBL, and 1.9% for VRE [30]. A re-examination in 2012 aimed to show possible changes in the prevalence and risk factors for MDROs. There was a significant decrease in cases of MDROs among the staff and residents of the geriatric care institutions (10.5 versus 53.8%), with only geriatric patients showing little change (22.7%) [46]. After another four years, March et al. [47] investigated the same geriatric care institution and geriatric ward in 2016, to obtain a trend for MDROs. The results indicated higher rates than in 2012 across all groups. Among the staff, a prevalence of 19.4% MDROs was identified, 11.9% of which was attributable to ESBL and 7.4% to MRSA. No staff member was colonized with VRE, MBL, and AmpC. Among residents and patients, a total of 66.1% and 26.0% positive cases were found, with cases positive for ESBL producers the most common. 

To investigate the prevalence and risk factors for MDROs, patients, residents, and staff were tested in three geriatric clinics, 40 geriatric care institutions, and two outpatient nursing care services in Frankfurt am Main. Samples were taken via nose, throat, and rectal swabs. MDROs were found in 6.3% of 64 staff members and in 20.1% of 288 residents and were attributed to MRSA, VRE, and ESBL-producing *Enterobacteriaceae*. No further characteristics of the MDROs or risk factors for staff can be obtained from this publication [13]. In a study in Taiwan, nasal swabs were taken to investigate *A. baumannii* and *S. aureus* presence in four geriatric care institutions and one cooperating hospital. Those tested included residents who spent the study period in the hospital and other residents, as well as the staff. The investigation was set up as a prospective study in order to determine persistent carriers. For the staff, it appears that a point prevalence study was carried out, but this is not explicitly reported. In this study, 53.8% of the 13 staff members tested positive for *A. baumannii* via nasal swab, but they did not show multidrug-resistance [32].

## 4. Discussion

This systematic review addressed the literature on the occupational risk of infection with multidrug-resistant organisms in non-outbreak situations for employees in the health service for the first time. The analyzed 22 studies were primarily cross-sectional studies in hospital or geriatric care settings with major differences in the number of participants. There was heterogeneity in the testing methods via swabs on various parts of the body and with various materials, the presence of a control group, and the MDROs investigated, as well as their prevalence. The most common pathogens were ESBL producers, VRE, and MRSA. 

Overall, there were only very few studies that defined the presence and prevalence of MDROs in employees as their study objectives. Patients or residents were often the main target group, and employees were only included incidentally in the investigations. Data on response rate, the execution of the studies for staff and medical professionals included were often missing or not available in sufficient detail. The study quality was rated as high in just two studies and as moderate in nine studies. Suitable control groups of non-medical staff or staff without patient contact were only included in three studies. These studies showed a higher risk of VRE for medical personnel and for those with patient contact versus staff without patient contact (9.6 versus 0%) [34] and for ESBL (4.0 versus 2.9%) [27]. On the other hand, the risk of MDR-GNB was shown to be higher among non-medical staff in the study by Liu [28] (15.0 versus 8.0%). However, this study only had a small control group of 133 people versus 1185 medical staff members, which was reflected in the update of this study (10.7%/1296 versus 12.9%/155) [45]. 

Unfortunately, the comparability of the studies was highly limited as a result of their heterogeneity. Any comparison of studies carried out in Asia, Africa, or South America with those carried out in Europe and North America should be viewed with caution, because of the prevalence of individual MDROs, as well as the incomparability of healthcare systems and hygiene standards. A differentiation into similar country groups was not possible as a result of the low number of studies included in the review. In addition to regional differences, there were also major variations in the number of participants, which, in 5 out of 16 studies for the hospital setting and in all studies in the geriatric care setting, amounted to fewer than 100 staff members.

### 4.1. MDROs

Previously, the prevalence of ESBL producers had primarily been investigated in patients and the general population. A prospective cohort study investigated nasal colonization in the general population in two regions of Germany (North Rhine-Westphalia and Lower Saxony) (*n* = 1878) and, despite a high *Enterobacteriaceae* rate of 33%, no cases of ESBL were detected [7]. However, another German study from Bavaria conducted among 3344 people with close contact to patients with bacterial gastroenteritis showed a faecal colonization rate of 6.3% with ESBL-producing *E. coli* [48]. Meyer et al. [49] tested conference participants for VRE and ESBL and found no cases of VRE but a rectal ESBL colonization of 3.5%. Travel to Greece or Africa, as well as contact with pets, were identified as risk factors. Travel is described as a risk factor for ESBL producers in many studies [50,51], and this is also corroborated by the included study of medical students with clinical assignments outside of Sweden [39]. An ESBL producers’ prevalence of 5.95% was observed among veterinary staff in the UK, which reflected a similar rate in the general population. In a longitudinal study over a period of six weeks, however, 25.9% of participants showed at least one positive result. The authors would therefore like to draw the attention of the healthcare providers to veterinary staff as a high-risk group [52]. The prevalence of ESBL producers in this review was between 3.5% and 7.1% for studies from Europe and the US with a moderate quality rating [27,38,39].

A current cohort study on VRE colonization has shown that it is possible for pathogens to be transmitted from patients to the gloves and gowns of medical staff, in particular from patients with higher bacterial burden [53]. The topic of hand hygiene has been addressed in numerous other studies via hand swabs, because hand hygiene plays a major role in the prevention of MDROs. This review therefore excluded all studies where the prevalence of MDRO was only determined using hand swabs from employees. In a current review, there was a pooled prevalence of MRSA in cross-sectional studies of 3.3% (*n* = 31): 8.3% for North America and 2.5% for Europe. VRE was not tested as often via hand swabs, and the prevalence here was 9.0% (*n* = 8). For MDR-GNB, there was a pooled prevalence of 4.6% for *Pseudomonas aeruginosa* (*n* = 17) and 6.2% for *A. baumanii* (*n* = 14). This review also stated the variation of region, study design, setting and sampling, and frequently missing information. One limitation of the review described by the authors is that the studies took place in a period of over 30 years, because the prevalence rates and testing methods have changed during this time [54].

### 4.2. Limitations

The limitations of this review are primarily based on the heterogeneity of the studies leading to the variation in the results. This can be seen, among others, in the MDROs investigated, screening method, sample size, and inclusion of a suitable control group. This severely limits the comparability of the studies and makes it difficult to draw a clear conclusion with regard to occupational infection risks. Another problem is the frequent missing of information about the participating staff in the studies. The study objective was often focussed on collecting information on MDROs in patients, and this is described in detail. The staff members tested in this context were often few in number, and the information on their acquisition and the description of the exposure were insufficient. Another limitation was the lack of differentiation between transient and persistent colonization with MDROs; this differentiation was not possible because of the commonly used cross-sectional study design in which one-day prevalence rates were collected.

## 5. Conclusions

This review presents the existing studies on the occupational infection risk with MDROs among employees in the health service. It shows that only very few, high-quality studies are available worldwide on MDROs in health personnel. Often, only a few employees were investigated at the same time as patients. Any comparison of the results is limited by their heterogeneity. More high-quality research is needed in order to describe the occupational infection risk with MDROs. In the investigations, for example, a sufficient number of study participants, the inclusion of an appropriate comparison group, a detailed description of the survey, and a clear and comprehensible presentation of the results are required.

## Figures and Tables

**Figure 1 ijerph-16-01983-f001:**
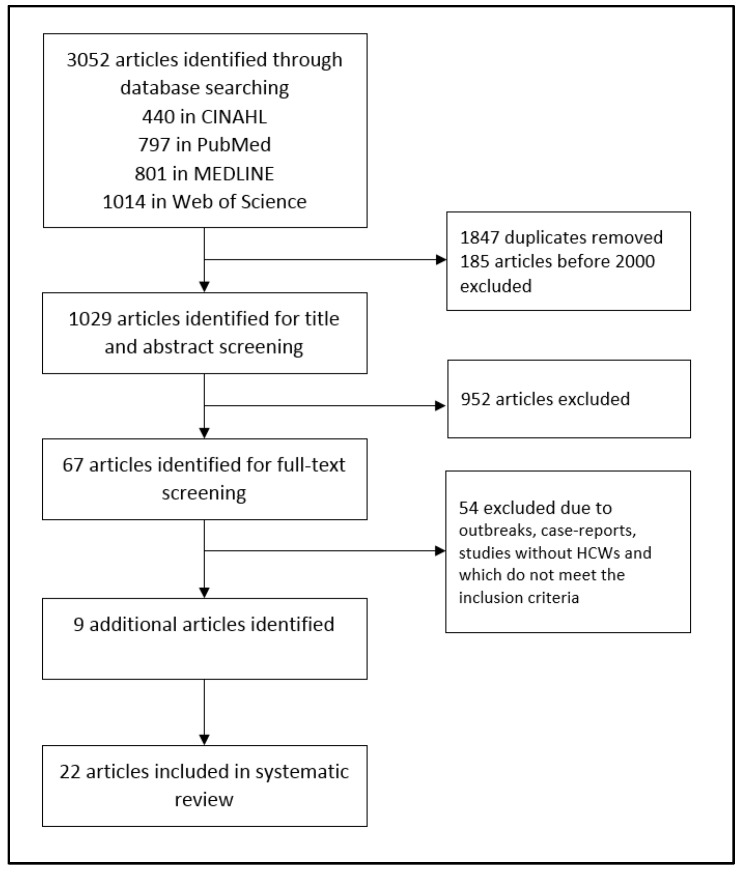
Flow diagram of study selection.

**Figure 2 ijerph-16-01983-f002:**
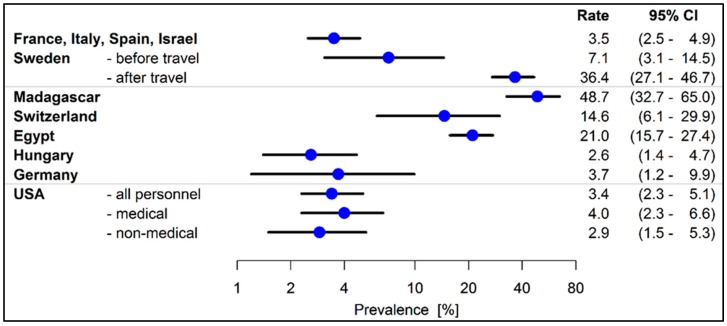
Extended-spectrum ß-lactamase (ESBL) producers’ prevalence among hospital employees.

**Table 1 ijerph-16-01983-t001:** Studies on multi-drug-resistant organisms (MDROs) in hospital employee.

	Country	Study Design	Study Period	Setting	Sample Type	Sample Size	MDRO Prevalence	Quality Assessment
Adler 2014 [38]	France (F), Italy (I), Spain (E), Israel (IL)	prospective	2008–11	rehabilitation units	rectal	1001 F 147 I 439 E 47 IL 368	ESBL 35 (3.5%) 3 (2.0%) 12 (2.7%) 5 (10.6%) 15 (4.1%)	++
Angelin 2015 [39]	Sweden	prospective	2010–14	abroad clinical assignment	faecal	99	ESBL before 7 (7.1%) ESBL after 36 (36.4%) CPO 0%	++
Moraes 2014 [40]	Brazil	retrospective	2005–12	hospital	different samples	105	MDRO 0%	++
Baran 2002 [34]	USA	cross-sectional	1998	hospital	stool	52 MP 40 NMP	VRE 5 (9.6%) MP VRE 0% NMP	++
Bayuga 2002 [33]	USA	cross-sectional	2000–01	hospital	nasal, hand	184	*Acinetobacte baumannii* 1 (0.5%)	++
Kocak Tufan 2010 [41]	Turkey	cross-sectional	2005	hospital	stool	221	VRE 0%	++
Andriatahina 2010 [42]	Madagascar	(cohort study) cross-sectional	2008	hospital	rectal	39	ESBL 19 (48.7%)	+
Agostinho 2013 [43]	Switzerland	cross-sectional	2010–11	hospital	anal	41	ESBL 6 (14.6%)	+
Bassyouni 2015 [31]	Egypt	cross-sectional	2013	hospital	stool	200	ESBL 42 (21.0%) AmpC 6 (3.0%)	+
Leao-Vasconelos 2015 [36]	Brazil	cross-sectional	2009–10	hospital	saliva	294	MDRO 27 (9.2%)	+
Ebrahimi 2016 [18]	Hungary	cross-sectional	2010–13	hospital	faecal	424	ESBL 11 (2.6%)	+
Bitterman 2016 [44]	Israel	cross-sectional	2015	hospital	rectal	177	CPO 0%	++
Jozsa 2017 [35]	Germany	cross-sectional	2013–14	unspecified	rectal	107	ESBL 4 (3.7%) MRSA 0% VRE 0%	+
Liu 2017 [28]	China	cross-sectional	2007–15	hospital	nasal, hand	1185 MP 133 NMP	MDR-GNB 104 (8.8) MP MDR-GNB 20 (15.0) NMP MRSA 11 (0.9%) MP VRE 1 (0.1%) MP	+++
Decker 2017 [27]	USA	cross-sectional	2013–15	hospital	perirectal	379 MP 376 NMP	VRE 0% MP+ staff ESBL 15 (4.0%) MP ESBL 11 (2.9%) NMP CPO 0% MP CPO 1 (0.3%) NMP	+++
Ake 2011 [37]	USA/Iraq	surveillance	2007	hospital	groin, axillar	80	MDRO 3 (3.8%)	++

CPO: carbapenemase producing organism, MP: medical personnel or personnel with patient contact, NMP: non-medical personnel or without patient contact, MRSA: methicillin-resistant *Staphylococcus aureus*, VRE: vancomycin-resistant enterococci, MDR-GNB: multidrug-resistant Gram-negative bacteria. Quality assessment: +++ high, ++ moderate, and + low study quality.

**Table 2 ijerph-16-01983-t002:** Studies on MDROs in employees in geriatric care.

	Country	Study Design	Study Period	Setting	Sample Type	Sample Size	MDRO Prevalence	Quality Assessment
Aschbacher 2016 [29]	Italy	review	2000–16	LTCF	nasal, rectal, inguinal, oro-pharyngeal, urine		MRSA 5.8–14.5% ESBL 5.2–7.0% CPO 0–1.5% VRE 0%	not applicable
March 2010 [30]	Italy	cross-sectional	2008	LTCF, geriatric unit	nasal, rectal, inguinal, oropharyngeal, urine	69	MRSA 10 (14.5%) ESBL 10 (14.5%) VRE 0% MBL 1 (1.5%) AmpC 1 (1.5%)	+
March 2014 [46]	Italy	cross-sectional	2012	LTCF, geriatric unit	nasal, rectal, inguinal, oropharyngeal, urine	57	MRSA 4 (7.0%) ESBL 3 (5.3%) MBL 0%	+
March 2017 [47]	Italy	cross-sectional	2016	LTCF, geriatric unit	nasal, rectal, inguinal, oropharyngeal, urine	67	MRSA 5 (7.4%) ESBL 8 (11.9%) VRE 0% MBL 0%	+
Gruber 2013 [13]	Germany	cross-sectional	2006–07	nursing homes, geriatric clinics	nasal, throat, rectal	64	MRSA 2 (3.1%) ESBL 2 (3.1%) VRE 1 (1.6%)	++
Liou 2017 [32]	Taiwan	(prospective) cross-sectional	2014–16	LTFCs, hospital	nasal	13	*A. baumannii* 0%	+

LTCF: long-term-care facility, MBL: metallo-ß-lactamase. Quality assessment: +++ high, ++ moderate and + low study quality.

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
