# Peer review of "Occupational Infection Risk with Multidrug-Resistant Organisms in Health Personnel—A Systematic Review"

_ijerph, 2019, doi:10.3390/ijerph16111983_

Round 1

Reviewer 1 Report

This systematic review is well written to summarize the literature on the occupational risk posed by multidrug resistant organisms in non-outbreak situations for employees in the health service. The well summarized tables are useful for readers including current information. The limitations of this review are primarily the result of the heterogeneity of the studies. However, this review may stimulate further high-quality and collaborative multicentre research in the field.

Given travelling to South-East Asia is an important risk factor, studies undertaken in South-East Asia (e.g. Indian) could be valuable to compare and contrast. But it seems authors have not shown any studies undertaken in South-East Asia (e.g. Indian). Is there any specific reason to exclude those studies?

Author Response

Thank you for your comment. We have included all appropriate studies worldwide in the review through the systematic and manual search described above. There was no exclusion of studies on MRE from staff from any country or region, not even from South-East Asia. Only studies focusing on hand or environmental screening and MRSA were not included.

We had the English language checked once again by an expert.

Reviewer 2 Report

In this review, the problems associated with occupational risk posed by multidrug-resistant organisms in non-outbreak situations for employees in the health service are addressed. The work follows a literature-review approach, including the analysis of 22 cases were primarily cross-sectional studies in hospital or geriatric care settings are presented. Even when the studies are spread over a diverse occupational risk, with different methodology and statistical data are far to be of confidence, it is clear that the most common pathogens were ESBL producers, VRE and MRSA.

The authors conclude that "more research is required in order to describe the occupational infection risk with MDROs", which is probably an obvious conclusion already present in the Introduction.

Thus, the article is of interest and the topic is well suited for the broad audience of IJERPH.

As the major correction and based on the analysis of the 22 cases here studied, I would like to suggest the author to go ahead and to propose a few concrete line of action to follow in order to bridging the gap between the current state of the art and the optimal studies necessary for determine the main issues associated with occupational risk posed by multidrug-resistant organisms.

Minor corrections, in the Abstract, please indicate the meaning of MRSA and VRE.

Author Response

Thank you for the good advice, which we were pleased to implement. We have added a few suggestions for good study quality into the conclusion. We emphasize the importance of a sufficient number of participants, a suitable comparison group and a detailed description of the study and the results.

We have included the changes to MRSA and VRE in the Abstract.